# Insight into the Behavior of Mortars Containing Glass Powder: An Artificial Neural Network Analysis Approach to Classify the Hydration Modes

**DOI:** 10.3390/ma16030943

**Published:** 2023-01-19

**Authors:** Fouad Boukhelf, Daniel Lira Lopes Targino, Mohammed Hichem Benzaama, Lucas Feitosa de Albuquerque Lima Babadopulos, Yassine El Mendili

**Affiliations:** 1Builders Lab, Builders Ecole d’Ingénieurs, ComUE NU, 1 rue Pierre et Marie Curie, 146110 Epron, France; 2Graduate Program in Civil Engineering—Structures and Civil Construction (PEC), Department of Structural Engineering and Civil Construction (DEECC), Technology Center (CT), Federal University of Ceará (UFC), Bloco 733, Campus do Pici s/n, Fortaleza 60440-900, CE, Brazil; 3Institut de Recherche en Constructibilité IRC, Ecole Spéciale des Travaux Publics, 28 Avenue du Président Wilson, 94234 Cachan, France

**Keywords:** artificial neural network, data processing, cement, glass powder valorization, life cycle assessment

## Abstract

In this paper, an artificial neural network (ANN) model is proposed to predict the hydration process of a new alternative binder. This model overcomes the lack of input parameters of physical models, providing a realistic explanation with few inputs and fast calculations. Indeed, four mortars are studied based on ordinary Portland cement (CEM I), cement with limited environmental impact (CEM III), and glass powder (GP) as the cement substitution. These mortars are named CEM I + GP and CEM III + GP. The properties of the mortars are characterized, and their life cycle assessment (LCA) is established. Indeed, a decrease in porosity is observed at 90 days by 4.6%, 2.5%, 12.4%, and 7.9% compared to those of 3 days for CEMI, CEMIII, CEMI + GP, and CEMIII + GP, respectively. In addition, the use of GP allows for reducing the mechanical strength in the short term. At 90 days, CEMI + GP and CEMIII + GP present a decrease of about 28% and 57% in compressive strength compared to CEMI and CEMIII, respectively. Nevertheless, strength does not cease increasing with the curing time, due to the continuous pozzolanic reactions between Ca(OH)_2_ and silica contained in GP and slag present in CEMIII as demonstrated by the thermo-gravimetrical (TG) analysis. To summarize, CEMIII mortar provides similar performance compared to mortar with CEMI + GP in the long term. This can later be used in the construction sector and particularly in prefabricated structural elements. Moreover, the ANN model used to predict the heat of hydration provides a similar result compared to the experiment, with a resulting R² of 0.997, 0.968, 0.968, and 0.921 for CEMI, CEMIII, CEMI + GP, and CEMIII + GP, respectively, and allows for identifying the different hydration modes of the investigated mortars. The proposed ANN model will allow cement manufacturers to quickly identify the different hydration modes of new binders by using only the heat of hydration test as an input parameter.

## 1. Introduction

With a production of about 10 Gm^3^/year, concrete is the most important building material and is, therefore, responsible for some of the world’s environmental problems due to the production of more than 4 billion tons/year of cement, which requires production temperatures higher than 1450 °C [1,2]. Indeed, concrete is responsible for 4 to 8% of the world’s CO_2_ emissions at all stages of production with 50% of the emissions in the construction sector alone [3,4]. Among materials, only coal oil and gas are a more important source of greenhouse gases [5].

One of the reflections that have emerged in recent decades is the substitution of part of the cement by other alternative binders, the supplementary cementitious materials (SCMs), with similar characteristics to cement, especially the pozzolanic and the hydration reactivities. The reduction in cement manufacturing’s carbon footprint remains a challenge and will have a strong appeal in the coming years.

Once glass becomes waste, it is landfilled, which is not sustainable, because it does not decompose in the environment. The recovery of glass waste eliminates the unnecessary consumption of limited landfill areas and can reduce energy consumption, among others. As a result, it is necessary to seek solutions that are more environmentally friendly. One of the proposed solutions is to replace part of the Portland cement, the most polluting component of concrete, with wastes and by-products, while trying to keep similar mechanical characteristics. Several industrial by-products have been successfully used as SCMs, including silica fume (SF), ground granulated blast furnace slag (GGBFS), and fly ash (FA) [6,7,8]. These materials are used to create blended cements that can improve concrete’s early durability and/or long-term strength, workability, and cost.

Glass powder (GP), which is studied in this paper, could also be a relevant solution for the creation of less-CO_2_-emitting concrete due to its high pozzolanic properties [9,10,11,12]. Each year, millions of tons of glass waste are generated worldwide [13,14]. In addition to reducing the amount of cement in concrete, glass waste will be recycled. This could be an important step toward sustainable development [15,16]. Furthermore, glass powder provides additional performance in terms of workability [17], microstructure properties [18,19], mechanical strength performance in the long term [20,21,22], durability [23], and hygrothermal performance [16,24,25].

In addition, in the context of sustainable applications of industrial by-products in environmentally friendly cementitious binders, Life Cycle Analysis (LCA) is a suitable tool for assessing the environmental impacts of cement production and its associated supply chains [26]. Assessing a literature review of LCA for the concrete and cementitious materials, it is seen that the major pollutant component is ordinary Portland cement (OPC), mainly due to the clinker manufacturing process. Indeed, there are three main steps for the cement manufacturing (cf., Figure 1): (i) the pre-processing, where the raw materials are crushed and milled into a fine powder, to enter the preheater; (ii) the calcination, where temperatures greater than 1400 °C are achieved, forming the clinker; (iii) finally, the addition of gypsum to control the setting time followed by final grinding to transform the clinker into a fine-grained mixture (90% of the particles diameter are on the order of 10 microns).

The main steps of the LCA assessment are: (i) determining the scope and limitations of OPC production and its partial replacement by more environmentally friendly SCMs, evaluating greenhouse gas emissions, and energy and raw material consumption; (ii) selecting output and input data using literature reviews to obtain average values for each necessary parameter [27,28,29,30,31,32,33], (iii) using the input data in the evaluation of the environmental impact of the mixtures proposed in this study by preceding the interpolation, and (iv) finally, presenting the results in terms of the decrease or increase in greenhouse gas emissions and energy and raw material consumption. The regulatory parameters of the two cements used, among others, are presented in Table 1, which includes the two types of cement used in this study (CEM I and CEM III). Indeed, the mineral content of CEM I is mainly pure gypsum (CaSO_4_⋅2H_2_O) with a maximum content of 5%, which controls setting time. Regarding CEM III, it contains GGBFS with a content between 36 and 95%. The current LCA assessment will be divided into three types of raw materials (i.e., clinker, GGBFS, and GP).

To investigate the hydration kinetics of these alternative binders, several models have been proposed in the literature over recent decades. The hydration heat models are dependent on several variables, some of which are uncontrollable. Finding a credible model that accurately describes the hydration heat is a challenging task. Based on the literature, the simulation methods can be classified into three categories: (i) linear regression models [34], (ii) physical models [35], and (iii) intelligent computing models. The linear regression models are constructed by using the least-squares methods to fit experimental data under the assumption that there are linear (or nonlinear) and independent relationships between the heat of hydration and its influencing factors, i.e., cement compositions, hydration age, and cement fineness. In the work of Qin [34], the non-assumptive projection pursuit regression method has been applied to predict the hydration heat of Portland cement.

Concerning the physical models, several models have been proposed in the literature to predict cement hydration heat. Tahersima et al. [35] used a finite element model to predict the hydration heat in a concrete slab-on-grade floor with limestone blended cement. The results showed that the prediction accuracy of finite element results is about 15% more for the maximum temperature rise and 30% more for the peak time. However, the main difficulty of this model lies in the determination of the thermal properties of the material to feed the model. Furthermore, the calculation time takes longer due to discretization. Kondo et al. [36] proposed a single-particle model that considers the circular-shaped layered growth of a uniform thickness of hydrates on a spherical single-reaction cement particle to characterize the hydration kinetics of alite (C_3_S). Pommersheim et al. [37,38] suggested an integrated single-particle reaction-diffusion model for the hydration of C_3_S. The hydration was modeled by using a classical approach to deal with reactions and diffusion at phase boundaries.

Recently, He et al. [39] developed a numerical method to describe the hydration heat characteristics of Portland cement under atmospheric steam curing conditions. This model has been employed using an iterative algorithm and confirmed with the experimental results. More recently, Nguyen et al. [40] investigated the effects of cement particle distribution on the hydration process of cement paste using a three-dimensional cement hydration computer simulation. The hydration process of cement paste was simulated using the model based on the CEMHYD3D code and the random distribution method was simpler and required less time in generating a pre-hydrated model than the correlation function method. In the work of Chu et al. [41], the CEMHYD3D code was also used to numerically model the hydration of cement paste by following the evolution of all phases and predicting the properties of hydrated cements. Furthermore, Zhao et al. [42] studied the early-age hydration of blended cement and modeled its hydration kinetics by measuring the hydration heat. Within this paper, the classification of hydration modes was discussed based on the Krstulovic–Dabic model [43,44], which was built on the data from the hydration rate. The limitations of numerical methods listed above indicate that there is a knowledge gap that needs to be filled, particularly for machine learning methods.

Indeed, intelligent computing models are usually based on artificial neural networks (ANN) and can also be used to assess the complicated process [45,46,47,48]. Nevertheless, a lack of knowledge on ANN applications to predict the hydration process was detected. Based on a literature review, limited works on Machine Learning (ML) and ANN applications applied to thermal data analysis and processing have been found in the literature. In addition, only two studies highlight the use of ANN models to investigate the hydration heat of Portland cement [49,50]. Subasi et al. [50] used a new approach based on an adaptive neuro-fuzzy inference system (ANFIS). Cook et al. [49] used a Random Forest (RF) model to achieve high-accuracy predictions of time-dependent thermal hydration kinetics of OPC-based analysis. Indeed, the largest drawback of RF is that it can become too slow and ineffective for real-time forecasts if there are too many combined decision trees. In general, these algorithms are quick to learn but take a long time to produce predictions after they have been trained. Moreover, ANFIS has a significant processing cost due to its complicated structure and gradient learning. This is a major drawback for applications with a high amount of data [51].

Due to the wide range of factors that influence the performance of the binder hydration process, the dynamic simulation of the hydration is a difficult task. Several variables influence heat hydration and they can occasionally be beyond control. It can be difficult to find a model that accurately describes the behavior of materials as they hydrate. Indeed, complex and non-linear systems influenced by their environment define behavior dynamics. However, the identification of the thermal parameters of the material to feed the model is where the main challenge of the physical models is presented in the literature. In addition, the discretization lengthens calculation times. For this reason, in this paper, an ANN model was used to predict the hydration process. The general objective is to fill the information gap and to present a dynamic model that can overcome the mentioned obstacles, compared to physical models, such as providing a realistic explanation of the heat of hydration, with minimum inputs and faster calculations. In order to contribute on that matter, the aims of this work were: (i) to assess the environmental impact of binders used and highlight the potential of glass powder (GP) valorization in the reduction in cement’s carbon footprint; (ii) to investigate the effect of GP on the hydration heat, and on the microstructural and mechanical properties of mortars, in addition to the thermogravimetric analysis results; (iii) to propose a numerical approach to identify and classify the transition modes of the cement hydration process. For this last purpose, a multilayer perceptron regressor (MLP) was proposed to generate a smooth curve model adjusted to the data reading signal and to map each experimental dataset. This method provides an interpolated model with less noise and/or other interferences. Once validated, the numerical analysis of the generated model was used to identify the transition modes. This identification was based on local maxima and minima of the generated shape format [52,53,54,55]. This method allows the development of a reliable model that can identify and set the hydration process [56].

## 2. Materials and Methods

Following the approach presented in Figure 2, the methodology employed in this paper is based on experimental and numerical steps. Each phase is made up of multiple sub-steps, as detailed in the following subsections. The first step concerns the materials characterizations at the laboratory scale and their carbon footprint assessment. The second step focuses on the data processing aspects and the general proposed numerical model dedicated to assessing the hydration process of the mortars used.

### 2.1. Experimental Program

Experiments were carried out to investigate the substitution effect of cement, which is responsible for significant greenhouse gas (GHG) emissions, by other non-biodegradable GHG alternatives, such as glass powder (GP), which represents a significant amount of waste. The use of GP as cement substitution would affect the consistency, the heat of hydration, and the mechanical properties. In the first step, a characterization of the binders’ constituents was performed through particle size analysis and density analysis. Afterward, mortars with Portland cement (CEM I 52.5 N), blast furnace cement (CEM III 32.5 N), and glass powder (GP) were mixed. In this study, 50% of the cement mass was replaced by GP. Indeed, Zeybek et al. [57] conducted a series of compressive strength, splitting tensile strength, and flexural strength tests to investigate the effect of waste glass powder on the mechanical properties of concrete. In their study, glass powder was used to replace up to 50% of cement. The study conducted by Kalakada et al. [58] showed also a decrease in compressive strength at high cement substitution. The compression strength of the concrete specimen was reduced by 65% when 50% of the cement was replaced with glass powder. In this study, it was chosen to replace the cement by 50% of GP in mass in view of other experiences in the literature with high substitution rates, but it is important to notice that focus is given on the proper validation of the relevance of the ANN model with materials having a low hydration heat (which is expected at those high substitution rates—up to 50%). The microstructural characterization of the tested mortars was conducted using an isothermal calorimeter, water porosity, and a TG analysis. In addition, the workability and mechanical strength of the mortars were also assessed and compared to control mortars with CEM I and CEM III. To explain the improvement of the mechanical strength and the decrease in the porosity in the long term, the thermogravimetric analysis was performed after 90 days.

#### 2.1.1. Raw Materials

Two types of cement were used in this study, CEM I 52.5 N and CEM III 32.5 N in compliance with the European standard [59]. Glass powder was used as mass substituents of cement. The glass powder was recovered from glass waste of demolition, provided by the waste disposal plant of Caen’s metropolitan region (Normandy, France). The density of binders was determined using a helium pycnometer and the particle size distribution was assessed by a laser particle size analyzer in the wet process in compliance with the European standard [60]. Figure 3 shows the particle size distribution of the used binders. The results show that the studied CEM I and CEM III were almost the same size as the literature result [61] and presented a finer particle size when compared to GP. Indeed, the average particle sizes D_50_ of the investigated binders were 16, 13, and 40 µm for CEM I, CEM III, and GP, respectively, as reported in Table 2. Several studies proved that a GP fineness finer than 75 µm would produce a pozzolanic reaction instead of alkali-silica reaction (ASR) [62,63]. Concerning the density results, CEM I presented a high density of about 3150 kg/m^3^ compared to those of CEM III and GP, which were 2880 and 2510 kg/m^3^, respectively.

#### 2.1.2. Mix Design and Experimental Protocol

The mortar mixture was prepared using the mix design method described by Equations (1)–(4) [64].
(1)$$V_{S}+V_{C}+V_{GP}+V_{W}\;\approx \;1\;m^{3}$$
(2)$$\frac{W}{GP+C}=0.5$$
(3)$$\frac{GP}{C+GP}=0.5$$
(4)$$\frac{S}{GP+C}=0.3$$
where *V_S_*, *V_C_*, *V_GP_*, and *V_W_* (m^3^) are the sand, cement, glass powder, and water volumes, respectively. *S*, *W*, *GP*, and *C* (kg) are the sand, effective water, glass powder, and cement contents per cubic meter of mortar, respectively. The water-to-binder ratio was taken as equal to 0.5 (Equation (2)). Such values of paste volume and water-to-binder ratio were chosen to obtain a homogeneous mortar mixture by limiting bleeding and segregation and to ensure its good workability [64]. Moreover, the water-to-binder ratio was chosen high enough to mitigate the phenomenon of self-desiccation due to hydration [65,66]. The cement substitution ratio was fixed at 0.5 (Equation (3)) to maintain the glass powder content (*GP*) as equal to the cement content (*C*). The volume of sand (*V_S_*) was fixed to obtain a total mortar volume of 1 m^3^. The mortar mix compositions are given in Table 3. A commercial SikaCem^®^ superplasticizer polycarboxylate was added to all the mixtures in order to obtain a better and comparable consistency. Indeed, due to the low content of the superplasticizer, compared to the mortar constituents, its content was neglected. The superplasticizer/cement ratio varied between 0.36% and 0.54%. The slump measured at the fresh state varied between 12 and 21 mm.

In the fresh state, the workability of mortars was assessed using a flow table test according to the ASTM standard [67]. This test is performed by placing mortar in a mini mortar cone, which has been previously wetted, in two layers on a vibrating table. Then, 15 vibrations are applied with a speed of 1 vibration/second. At the end of this test, the flow is measured. In the hardened state, the porosity, and the compressive and the tensile strengths of all specimens were tested. According to the French standard [68], the prismatic mortars of 4 × 4 × 16 cm^3^ were cast. The specimens were cured at the standard conditions, i.e., for 24 h at 20 ± 1 °C and relative humidity no less than 60%. Then, the specimens were unmolded and cured in water for different test ages (3, 14, 28, and 90 days) at (22 ± 1) °C. After that, all mortar specimens were taken out of the water for the mechanical properties tests.

The test of the water porosity consisted of placing the specimens in saturation benches until the total saturation in water, which was ensured by a vacuum pump [69]. Following 24 h of saturation, the specimens were collected, wiped, and their wet and underwater weights measured. After that, the specimens were put in the oven at 105 °C for 24 h and a second dry weight measurement was taken. The porosity corresponded to [69].

Concerning the mechanical tests, the corresponding loading rates of the compressive and flexural strength were 2.4 ± 0.2 kN/s and 50 ± 10 kN/s, respectively. Finally, the thermogravimetric analysis (TG) was carried out to monitor the solid phase formations on the mortars and the possible pozzolanic reaction (portlandite—CH, calcite—CC, etc.). The device used was the NETZSCH^®^ instrument (STA 449 F5 Jupiter, Selb, Germany). It consists of a sealed chamber to control the atmosphere of the sample by injection of helium gas, an oven to manage the temperature, a weighing module (microbalance), a thermocouple to measure the temperature, and a computer to control and record data. During the test, the temperature should be increased from room temperature to 1200 °C with a constant rate of 10 °C/min. This velocity is widely used in the literature and it is recommended by the French standard [69]. The obtained TG results were used to calculate the portlandite rate (CH) and the chemically bound water rate (*w_b_*) by Equations (5) and (6) [70].
(5)$$CH\%=\frac{m_{410}-m_{490}}{m_{490}}\times \frac{M_{Ca{\left(OH\right)}_{2}}}{M_{H_{2}O}}\times 100\%$$
(6)$$w_{b}\%=\frac{m_{40}-m_{490}}{m_{490}}\times 100\%$$
where *m*_40_, *m*_410_, and *m*_490_ are the sample weights at 40 °C, 410 °C, and 490 °C, respectively. $M_{Ca{\left(OH\right)}_{2}}$ and $M_{H_{2}O}$ are the molecular masses of portlandite (CH) and water (H_2_O), respectively. To compare the results, *CH* rate and *w_b_* rate were normalized to 100 g of anhydrous material.

### 2.2. Data Processing Model

In this section, a binder hydration monitoring method is proposed. This method is based on an artificial neural network (ANN) model processing the data from the hydration heat tracking test. Indeed, a multilayer perceptron regression (MLP) was used to proceed to an initial treatment of the data considered as signals in order to transform them into a time series regression. Afterward, a numerical analysis of the generated model was carried out in order to identify the different transition modes. The identification of the transition modes was based on the local maxima and minima values of the generated model and related to the hydration process.

#### 2.2.1. Artificial Neural Network Modeling

In the literature, the most used data-driven model for building and material behavior is the artificial neural network (ANN) [71,72,73]. Numerous types of ANNs have been developed over the years with varying characteristics. A multilayer perceptron (MLP) is a class of ANN, which is applied for both classification and regression problems, it is a classical type of neural network, very flexible, and can be used generally to learn a map from inputs to outputs, among other applications. It is classified as supervised learning, using a backpropagation technique for training (cf., Figure 4). All studies in the literature highlight good results in long short-term memory (LSTM) applications showing versatility in a multitude of scenarios [74,75,76,77].

The MLP model consists of three parts or nodes organized in the input layer, hidden layers, and output layer. Except for the input nodes, all the other concepts apply non-linear activation functions in which the size of the hidden layer is usually determined by hyperparametric optimization [75]. The details of the inputs and internal weights are optimized depending on the applied problem and dataset.

As seen in Figure 4b, the neuronal structure is also composed of three elements [78,79]: (i) ‘synapses’ or connecting links that are associated with particular weights; (ii) the processing unit that handles the input signals weighted (summing junction) by the synaptic weight and adjusts them by adding a bias value (bk); (iii) an activation function (φ) that limits the signal amplitude at the neuron’s output. Figure 4b demonstrates a typical neuronal structure, applied for the hidden and output layer. The input layer differs by the absence of the activation function and the single input synapse, connected to the input data. The neuron k can be described mathematically by Equations (7) and (8).
(7)$$u_{k}=\overset{m}{\underset{j=1}{\sum }}x_{j}\times w_{kj}$$
(8)$$v_{k}=u_{k}+b_{k}$$
where *u_k_* is the linear combiner of the input signal and *v_k_* is the weighted sum of the input values adjusted by the *b_k_*. The activation function is a set of mathematical equations that define the output signal of the neuron k. The activation function may be divided into four distinct groups [80], as in Equations (9)–(12). Equation (9) represents the threshold function, which is equivalent to a Heaviside function. Equation (10) is an identification function (linear function). Equation (11) is a logistic sigmoid function (having a sigmoidal nonlinearity with a as its slope parameter). Equation (12) is a hyperbolic tangent function (another kind of nonlinearity).
(9)$$\varphi (v)=\left\{\begin{array}{c} 1\;if\;v>0\\ 0\;if\;v=0\\ -1\;if\;v<0 \end{array}\right.$$
(10)φ(V)=0
(11)$$\varphi (V)=\log f(V)=\frac{1}{1+e^{-\alpha v}}$$
(12)$$\varphi (V)=\tanh(V)=\frac{2}{1+e^{-\alpha v}}$$

The sum of the weighted and bias-corrected inputs passes through the activation function *φ* to obtain the final output signal (*y_k_*), as shown in Equation (13).
(13)$$y_{k}=\varphi (\overset{m}{\underset{j=1}{\sum }}x_{j}\times w_{kj}+b_{k})$$

The implementation of the ANN model is carried out in three steps. The initial stage is to train the model to minimize the absolute error by altering the weight parameters, based on comparisons between experimental and model outputs, which should be as close as possible. The LBFGS (Limited-Memory Broyden–Fletcher–Goldfarb–Shanno) is a popular optimization algorithm for estimating weight parameters in machine learning. It uses a quasi-Newton method [81,82,83]. As the original method (LBFGS), it uses an estimate of the inverse Hessian matrix, but in that case, LBFGS stores only some chosen vectors, to be able to represent the model with a required approximation.

#### 2.2.2. Evaluation Metrics

In order to evaluate the final model’s performance, the correlation coefficient (*R*²), the root mean square error (*RMSE*), and the mean absolute error (*MAE*) are employed [84,85,86]. The aim is to achieve a model that best fits the labeled training data by evaluating all the combined applied metrics. All these metrics are performance indicators commonly used to measure the efficiency of the machine learning models generated. These indicators’ definitions are given in Equations (14)–(16), respectively.
(14)$$R^{2}=1-\frac{\sum_{i=1}^{n}{\left(y_{i}-{\hat{y}}_{i}\right)}^{2}}{\sum_{i=1}^{n}{\left(y_{i}-\bar{y}\right)}^{2}}$$
(15)$$RMSE=\sqrt{\frac{1}{n}\times \overset{n}{\underset{i=1}{\sum }}{\left(\hat{yi}-yi\right)}^{2}}$$
(16)$$MAE=\overset{n}{\underset{i=1}{\sum }}\left|\frac{yi-\hat{y}}{yi}\right|$$

#### 2.2.3. Data Processing/Numerical Model

In addition to the characterization of the microstructure and mechanical properties at several ages, a conceptual framework was built to describe and visualize our study methods after reviewing the relevant current literature [42,44,45,46,47,48]. Each phase is composed of several steps, as explained in detail in the following sections. These main phases are summarized in Figure 5 and as below:Collecting data: the experimental data were collected based on heat energy measures within the time, during the cement hydration process. That was the base dataset used in the initial data processing, heat energy within time. Four datasets obtained from CEM I, CEM III, CEM I + GP and CEM III + GP were analyzed. Each dataset contained X data points, in a total of Y datapoints being analyzed;Model estimation: concept of the initial data processing, testing, and optimizing all the parameters necessary for the execution of the algorithm, such as MLP hidden layers size, activation functions, and optimization algorithm [71,72,82,85];Evaluation of conceived model: based on the mentioned metrics parameters (R², RMSE, MAE), the best parameters setup was evaluated to proceed with the data classification [85,86];Mode’s identification: based on the generated model data (hydration heat Q vs. time plot), a numerical peak analysis was proceeded to identify the transition zones and, thus, the transition modes of binder hydration with time [42].

As a result, the binder hydration modes were obtained, through a staircase levels plot, with respective time transitions [42,45,48]. In addition, within the chemical compositions, the main difference aspects can be highlighted in the behavior, and the total time of each mode for different samples can be compared.

## 3. Results and Discussion

### 3.1. Life Cycle Analysis LCA

Table 4, Table 5 and Table 6 present the LCA inventory obtained after interpolations and considerations reported in the literature. GGBFS is highlighted by the literature with a potential emission reduction of 78.84% in the production of CEM III [64]. Regarding GP production, its manufacturing process shown in Table 4 is not on an industrial scale according to [65]. For masses, units given are in kg of raw material by kg of produced binder (kg/kg product). For raw materials in volume, such as water, units given are in m^3^ of raw material by kg of produced binder (m^3^/kg product). For energy input, it is either given in terms of kWh of energy by kg of produced binder (kWh/kg product) or in kg of utilized fuel source by kg of produced binder (kg/kg product).

Figure 6 shows the different raw materials’ consumption. The results show a considerable reduction in natural resources replaced by waste. In general, this also implies a reduction in land use, extraction, and other environmental aspects related to the extraction of mineral raw materials using by-products, avoiding dumping and similar impacts.

Figure 7 presents the results for the relevant energy demands. Apart from the electricity produced by different renewable resources (hydroelectric, wind, and thermoelectric), all other resources are derived from fossil origins. The fuel that has the greatest impact on clinker production is petroleum coke and hard coal, followed by fuel oil. The proposed formulations with a 50% replacement of cement will contribute to a direct 50% reduction in emissions once the clinker manufacturing process is significantly reduced. Regarding the increase in natural gas consumption, LCA of GP reveals that the results presented are not applied on an industrial scale, while scalability in this area could increase performance by reducing consumption.

Figure 8 shows the GHG emission results for the formulations studied. The results show a significant reduction in GHG for the formulations containing GP. Regarding mainly CO_2_ emissions, CEM III + GP shows a low impact mainly due to the 78.83% reduction of the CEM III formulation and its 50% replacement by GP. Finally, the same reduction profile is observed for the case of heavy metals, i.e., nitrous oxide (NO_x_) and sulfur dioxide (SO_2_). According to the literature, NO_x_ is more detrimental to climate change than CO_2_, while some authors indicate that it could be 300 times more detrimental to the climate than CO_2_.

### 3.2. Experimental Results

Table 3 presents the flow values measured of the tested mortars. The flow values vary between 12 and 21 mm. Moreover, an improvement in workability is obtained of up to 41.6% and 23.5% for CEM I + GP and CEM III + GP compared to mortars with CEM I and CEM III, respectively. This result is attributed to the higher fineness of GP compared to cement. Therefore, the substitution of cement by GP decreases the water demand for hydration process, i.e., by increasing the amount of free water and workability of the pastes [24].

Figure 9 presents the evolution of the porosity, and the compressive and the tensile strengths (from flexural tests) at several curing ages (3, 14, 28, and 90 days). In general, CEM III presents less water porosity in comparison with CEM I. This phenomenon is explained by the presence of slag in CEM III, which leads to the pores clogging by creating other ranges of non-connected pores [87]. However, after 3 days of curing, the mortars containing GP show high porosity compared to those with only cement. This phenomenon is due to the fineness of GP used, which changes the microstructure and the pore-size distribution [15,24]. A proportional decrease in porosity is observed with increasing age for all mix designs. At 90 days, this decrease is of the order of 4.6%, 2.5%, 12.4%, and 7.9% for mortars based on CEM I, CEM III, CEM I + GP, and CEM III + GP, respectively, compared to the results at 3 days. This result seems to be in good agreement with those found previously by other authors [15,24,49,50].

When it comes to mechanical strength, for all mortars, an increase is observed and this is due to the continuous hydration of the binder with the curing time [22]. In addition, the use of GP with a replacement of 50% of the cement decreases the mechanical strength in the short term. This is due to a dilution effect, which is the immediate consequence of the substitution of a more reactive powder (cement) with a less reactive powder (GP) in the short term [12]. Nevertheless, the mechanical strength does not cease increasing with curing time [15,22,24]. This phenomenon is due to continuous pozzolanic reactions between portlandite (Ca(OH)_2_) and silica originating from GP, as highlighted by Idir et al. [55], in addition to the slag present in CEM III, as also demonstrated below by the TG analysis. After 90 days of curing time, the CEM I + GP presents a decrease of about 28% and 17.7% in compressive and tensile strength, respectively, compared to CEM I. For CEM III + GP, the decrease is about 57% and 56% in compressive and tensile strength, respectively, compared to CEM III. These results are in agreement with those reported by Adesina et al. [88]. Nevertheless, these values are adequate for the use of CEM I + GP in the construction sector and particularly in prefabricated structural elements.

To investigate the pozzolanic reaction of GP with portlandite (CH), a thermogravimetric (TG) analysis is conducted at 90 days. Figure 10 shows the three main phases formed in mortars with CEM I and CEM III. The mortar with CEM III has less portlandite and calcite (CC) amounts than CEM I mortar does. This is due to the activated pozzolanic reaction of the blast furnace slag present in the CEM III cement. Nevertheless, in the case of mortars with GP, the two peaks related to portlandite and calcite are reduced, showing the consumption of CH by GP to produce additional CSH hydrates [24]. An increase in the first peak is observed. Indeed, the three phases correspond generally to three processes according to [89,90]. The first band below 200 °C is due to the loss of free water as well as the dehydration of calcium silicate hydrate (CSH), ettringite (A_Ft_), calcium monosulfo-aluminate, hemicarboaluminate, or calcium monocarboaluminate (all these material phases are denoted as A_Fm_). The second peak around 400–500 °C is contributed to the dehydroxylation of portlandite (CH), while the decarbonization of calcium carbonates (CC) takes place at 700–800 °C. From dTG patterns, the A_Ft_ and A_Fm_ material phases are difficult to dissociate, as the corresponding bands are large, and the decomposition temperatures of these phases are very narrow and overlay each other.

Based on TG results, the normalized amounts of the bound water (w_b_) and portlandite (CH) are both presented in Figure 11. Indeed, the w_b_ rate indicates the volume of reaction products (including CSH, A_Ft_, A_Fm_, and CH) formed during cement hydration, while normalized CH can be interpreted as an indication of changes in the assemblage of reaction products (CH consumption due to the formation of additional CSH hydrates) as highlighted by Damidot et al. [91]. Figure 11 shows that for the mortars containing low clinker content, the w_b_ rate is lower than for the standard mortar (CEM I).

The increase in w_b_ in the mortars with GP is related to pozzolanic reactions. Indeed, silica reacts chemically with calcium hydroxide (CH) and produces additional CSH phases. Due to CH consumption, the normalized CH value for mortar with GP is lower than that for the standard mortars (CEM I and CEM III). These data show the same trend as those reported in literature [92,93], where w_b_ and CH values for both pure and blended pastes with silica fume were estimated and measured.

### 3.3. Data Processing/Numerical Results

Figure 12 shows the experimental data and those predicted by the ANN model for the heat of hydration in mW/g. Indeed, the models translating the hydration behavior of CEM I, CEM I + GP, CEM III, and CEM III + GP mortars provide a good agreement with the data found experimentally with a difference of relevance more important for the MAE and RMSE metrics. For the RMSE, the lowest results are obtained with the CEM III + GP results with a 0.92 R^2^ score, with the further metrics appointed in the same evaluation for near 0.35 for MAE, and with an improvement for the RMSE, at a value of 0.20, the best result for all datasets.

The hydration modes identification of CEM I, CEM I + GP, CEM III, and CEM III + GP are shown in Figure 13a,b and Figure 14a,b, respectively. In general, the proposed model identifies four main modes [32], including the initial reaction and slow reaction period (Mode 1), acceleration period (Mode 2), deceleration period (Mode 3), and stability period (Mode 4) in the history of heat flow. As shown in Figure 13 and Figure 14, the standard mortar with CEM I started an earlier acceleration period than the other mortars studied with a higher heat release. This mode corresponds to the induction phenomenon interpreted by the geochemistry dissolution theory that the primary mechanism controlling the kinetics up to the end of the induction period is the undersaturation of this surface layer [94]. In addition, tri-calcium aluminate (C_3_A) reacts promptly with water from an early age to form calcium hydroaluminate (3CaO–Al_2_O_3_–Ca(OH)_2_–nH_2_O or hydroxy–A_Fm_), as mentioned by Nguyen et al. [32].

As shown in Figure 13b and Figure 14a, CEP I + GP and CEM III mortars present almost the same behavior in term of peaks position. Nevertheless, these peaks start at 6.78 h and 7.58 h from the first water contact of the binder for the CEM III and CEM I + GP mortars, respectively. This time delay can be explained by the fact that the containing slag in the CEM III is more reactive than the glass powder.

In the second mode, the peak value decreases and shifts toward a longer hydration time when the CEM III and GP are used. This phenomenon could be attributed to the consumption of C_3_S (Alite), which is consistent with the hydration heat rate. It is known that the hydration rate of C_3_S is much higher than that of C_2_S (Belite) due to the higher solubility and lower activation energy of C_3_S compared to C_2_S [95]. Indeed, the solubility of C_3_S is between 5 and 8 times higher than that of C_2_S [95]. Therefore, it could be concluded that Mode 2 is strongly affected by the hydration of C_3_S, and this state is in good agreement with the literature [96]. At the end of the induction period, CSH and Portlandite (CH) begin to develop fast. There is still considerable controversy about the trigger for this increase in reaction rate. Indeed, in pastes, it seems that the CH precipitation is the trigger.

Mode 3 corresponds to the renewed dissolution of C_3_A with the formation of ettringite (A_Ft_). Due to this retardation of the C_3_A reaction with sulfate, the main reaction of this phase should occur after the main peak of the reaction of alite in a properly sulfated Portland cement. However, in CEM I, the formation of ettringite continues after the exhaustion of sulfate in the solution [97]. In conclusion, the ANN model makes it possible to determine when the initial hydration phase is completed, i.e., after 20.54 h, 42.40 h, 45.04 h, and 55.04 h for mortar based on CEM I, CEM III, CEM I + GP, and CEM III + GP, respectively.

## 4. Conclusions

An ANN model is used for the first time to identify the different hydration modes of binders using only the heat hydration test as an input parameter. The method is based on a MLP regressor that maps the input signal and generates a fitted model. Afterward, a numerical approach is proceeded to identify the different hydration modes with respective times.

Indeed, ANN model detects four modes during the hydration process. In addition, the model proposed provides similar results in terms of heat of hydration in comparison to the experimental results with R² = 0.997, 0.968, 0.968, and 0.921 for CEM I, CEM III, CEM I + GP, and CEM III + GP, respectively. In addition to the mechanical strength achieved, the replacement of 50% of cement by GP is significantly positive for the LCA results. Indeed, the GHG emissions present the most significant results, by reducing 50% and 80% of the emissions of CO_2_ and NO_x_ + SO_2_ for the CEM I + GP and CEM III + GP mixtures, respectively.

It is revealed in this study that waste glass can be employed as a partial cement substitute in mortar and would be beneficial for GHG emissions of concrete. The present study identifies waste glass as suitable, accessible in large quantities, a local eco-material, and suitable for concrete construction from an economic and environmental standpoint. Nevertheless, when waste glass is employed as partial replacement for cement, the strength values are reduced, which needs to be investigated in the lab for proper mix design and use of the materials. To address this shortcoming, future research will focus on combining waste glass with other recycling fibers. Moreover, the proposed ANN model allows cement manufacturers to quickly identify the different hydration modes of new binders by using only the heat hydration test as an input parameter.

## Figures and Tables

**Figure 1 materials-16-00943-f001:**
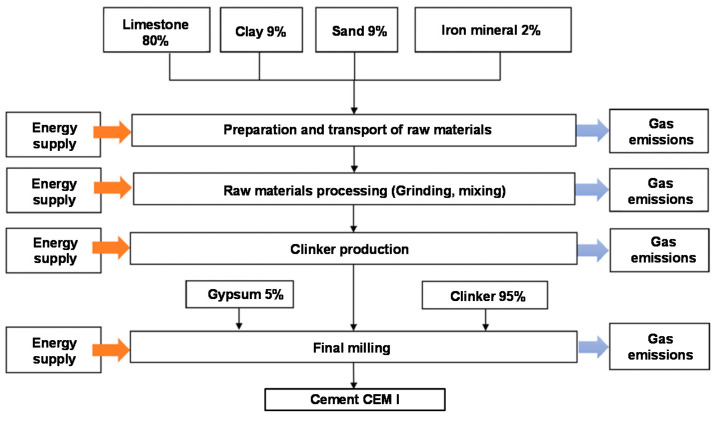
Basic process, “cradle-to-gate” scope and system limits [27].

**Figure 2 materials-16-00943-f002:**
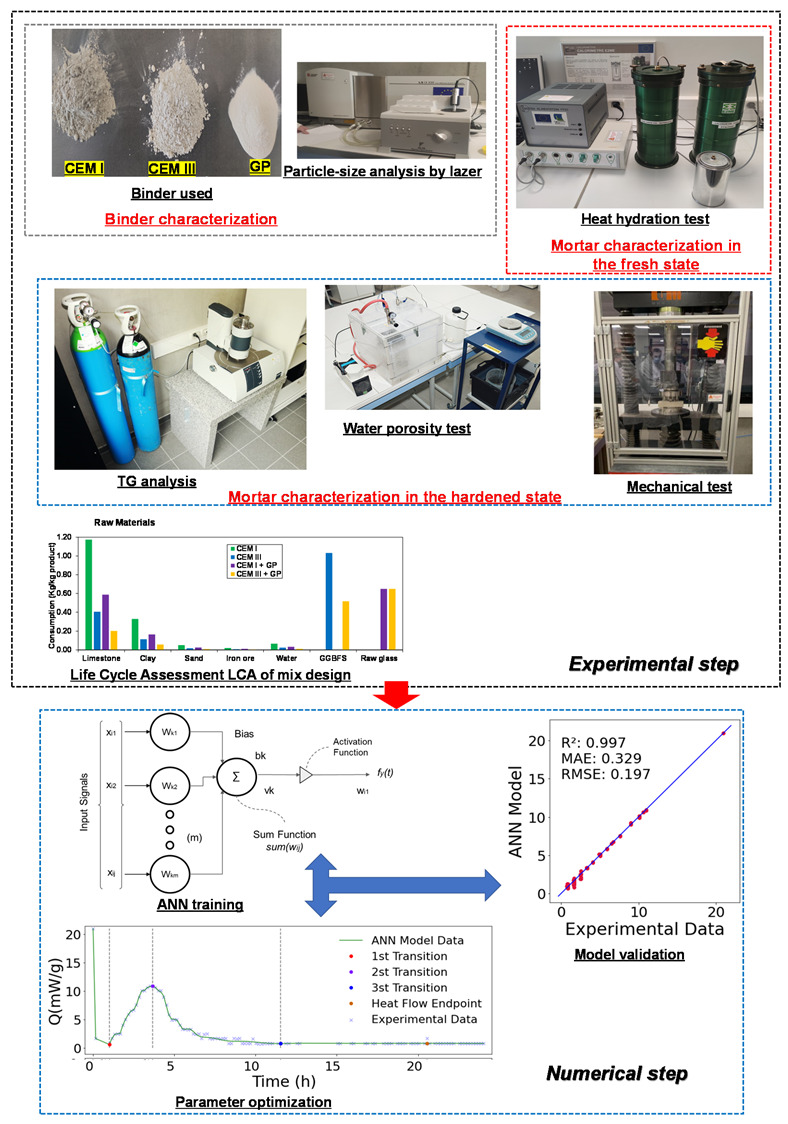
Conceptual study plan.

**Figure 3 materials-16-00943-f003:**
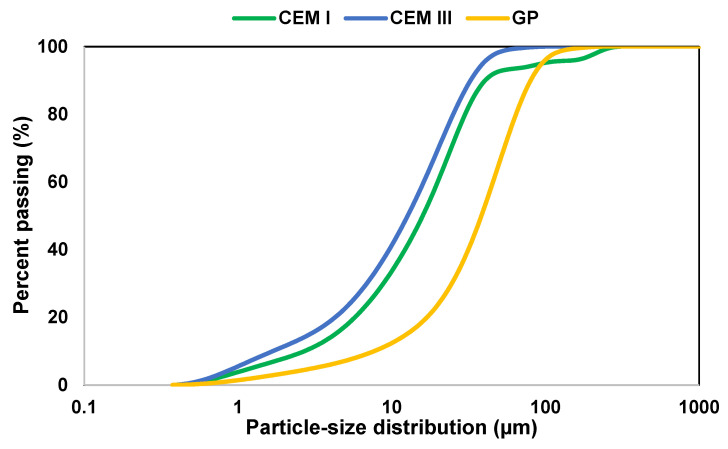
Particle-size distribution of the cements and glass powder used.

**Figure 4 materials-16-00943-f004:**
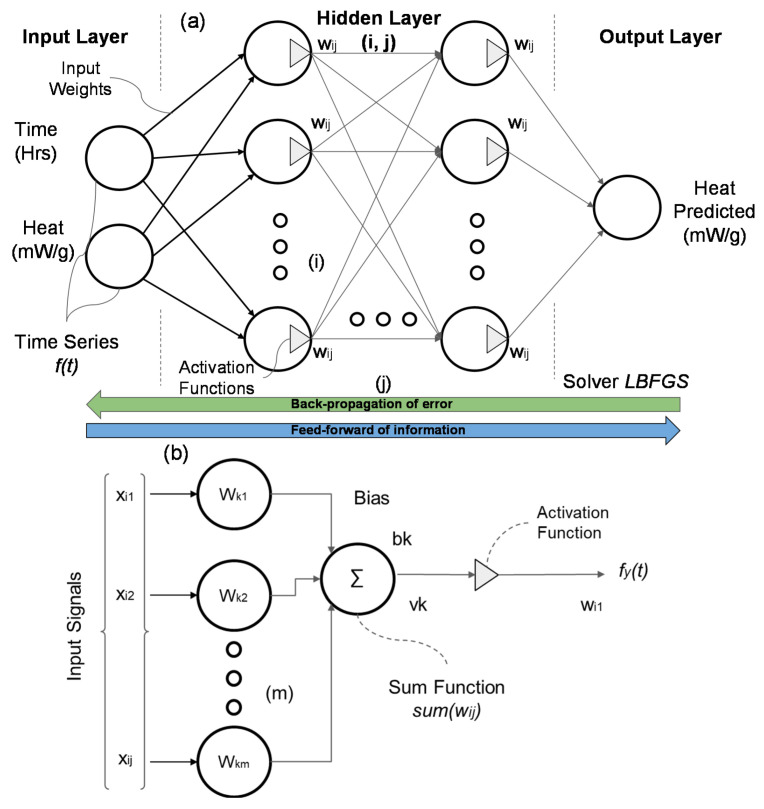
Simplified representation of an artificial neural network and its components: (**a**) Design Concept of a Multi-Layer Perceptron Model (MLP) and (**b**) Neuron simplified structure.

**Figure 5 materials-16-00943-f005:**
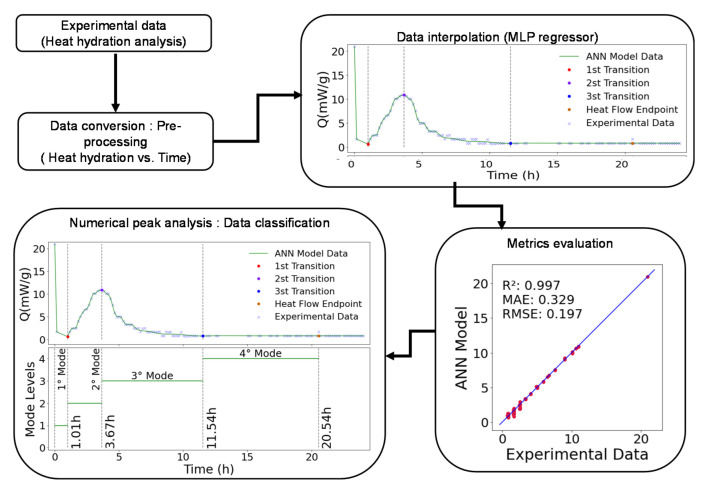
Computational procedure for hydration process classification proposed model.

**Figure 6 materials-16-00943-f006:**
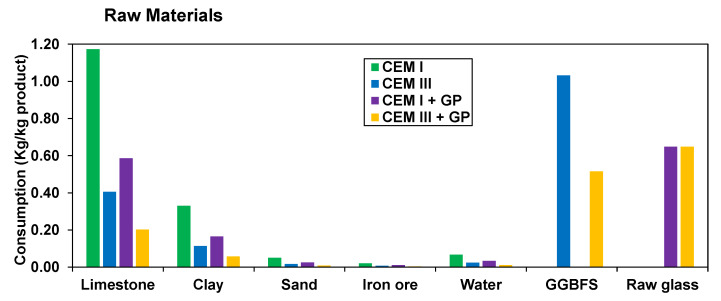
Raw materials’ consumption.

**Figure 7 materials-16-00943-f007:**
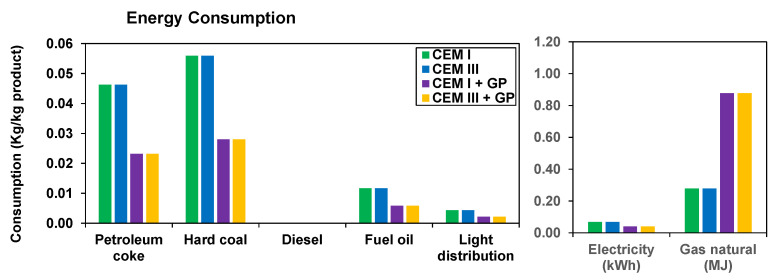
Energy consumption results.

**Figure 8 materials-16-00943-f008:**
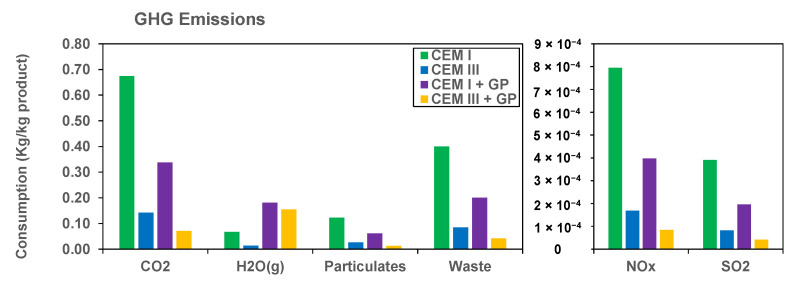
GHG Emissions results.

**Figure 9 materials-16-00943-f009:**
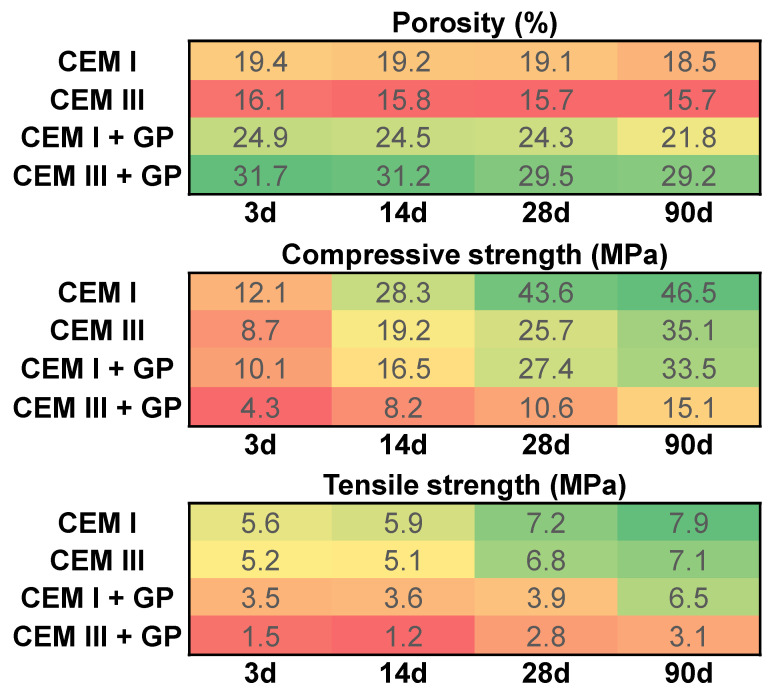
Porosity, and compressive and tensile strengths of mortars used.

**Figure 10 materials-16-00943-f010:**
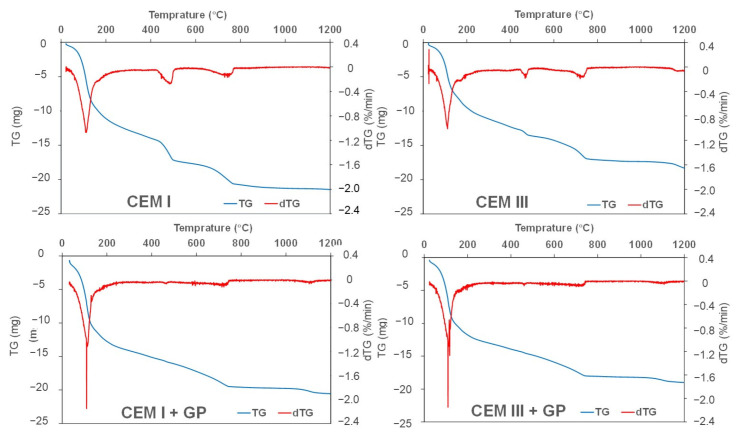
TG and dTG curves of the studied mortars.

**Figure 11 materials-16-00943-f011:**
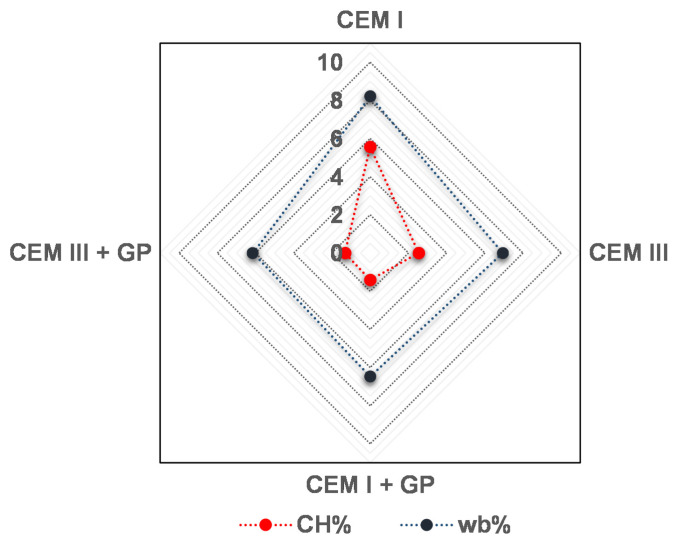
CH% and w_b_% of mortars studied.

**Figure 12 materials-16-00943-f012:**
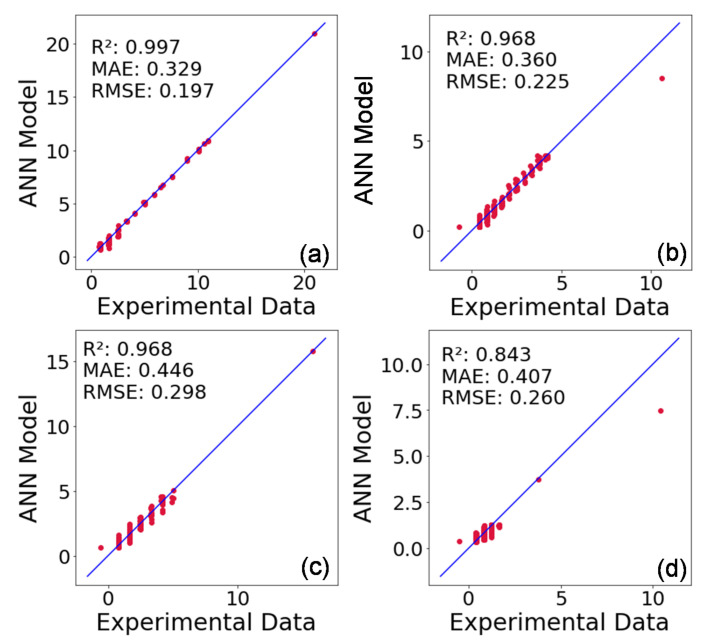
ANN models versus experimental data (for the heat of hydration in mW/g) with error metrics for (**a**) CEM I, (**b**) CEM I + GP, (**c**) CEM III, and (**d**) CEM III + GP.

**Figure 13 materials-16-00943-f013:**
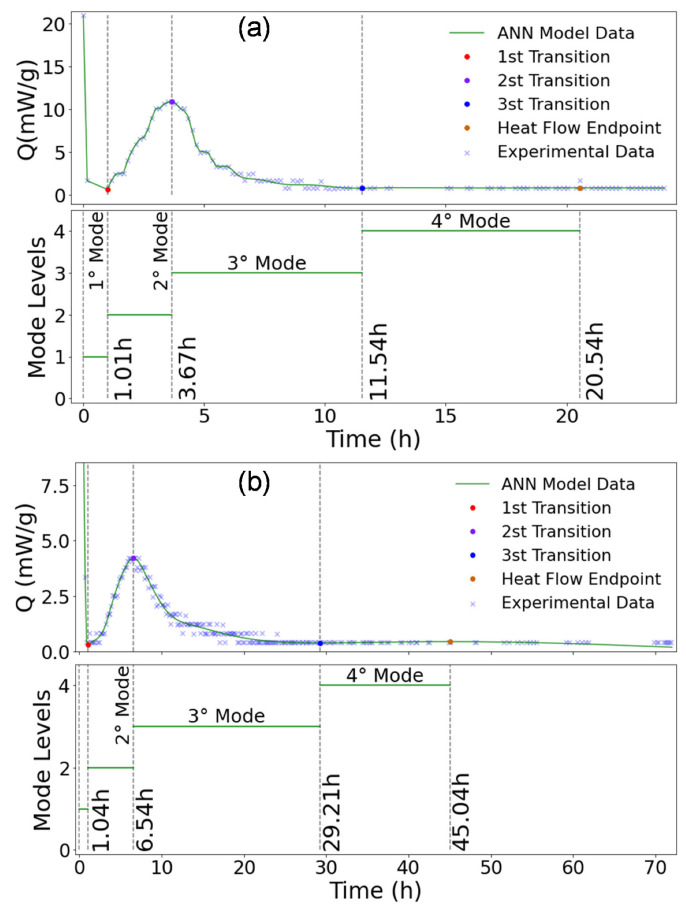
Heat of hydration, ANN model generated, modes, and transitions identification: (**a**) CEM I and (**b**) CEM I + GP.

**Figure 14 materials-16-00943-f014:**
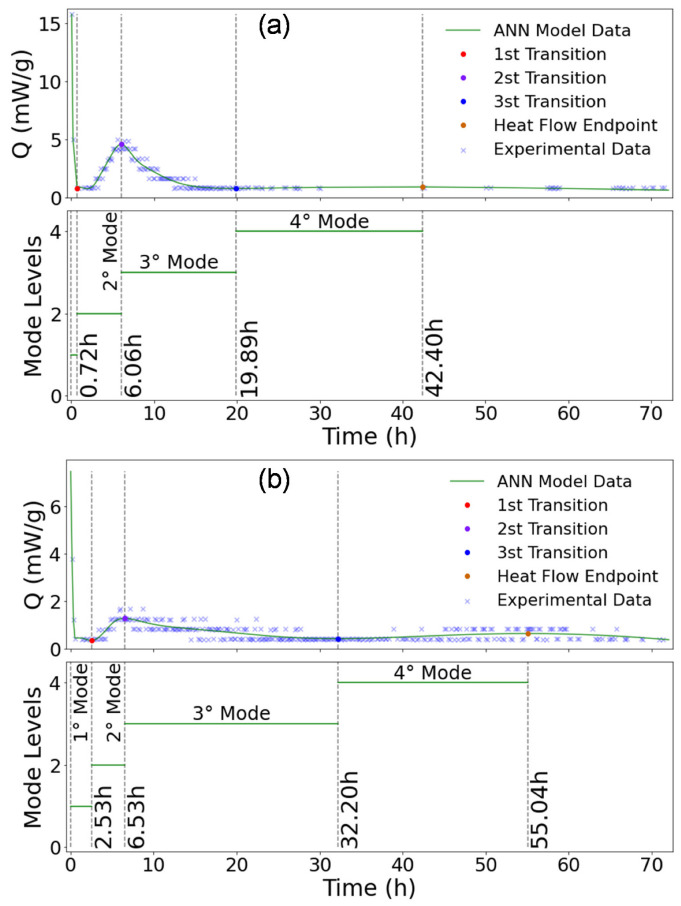
Heat of hydration, ANN model generated, modes, and transitions identification: (**a**) CEM III and (**b**) CEM III + GP.

**Table 1 materials-16-00943-t001:** Cement regulations following European Standard (EN 197-1) [27].

Class	Mineral Content		Mineral Components/Inorganic Process Additions	Market Share
	(Allowable Range)	(Average)		
Ordinary Portland cement (CEM I)	≤5%	2.5%	inorganic process addition	30%
Portland composite cement (CEM II)	6−35%	20.5%	ground granulated blast furnace slag (GGBFS), silica fume, pozzolan, fly ash, burnt shale	57%
Blast furnace cement (CEM III)	36−95%	65.5%	GGBFS	5%
Pozzolanic cement (CEM IV)	11−55%	33%	silica fume, pozzolan, fly ash	6%
Composite cement (CEM V)	36−80%	58%	GGBFS, pozzolan, fly ash	3%

**Table 2 materials-16-00943-t002:** Particles size and density of binders used.

	CEM I	CEM III	GP
d_10_ (μm)	3	1.5	8
d_50_ (μm)	16	13	40
d_90_ (μm)	40	32	80
d_max_ (μm)	300	70	120
Density (kg/m^3^)	3150	2880	2510

**Table 3 materials-16-00943-t003:** Mix proportions of mortar (kg/m^3^).

	CEM I	CEM III	CEM I + GP	CEM III + GP
Sand 0/4	1795	1638	1761	1685
CEM I	500	0	250	0
CEM III	0	500	0	250
GP	0	0	250	250
Water	250	250	250	250
Superplasticizer	1.8	1.5	1.4	1.2
Flow (cm)	12	13	17	21

**Table 4 materials-16-00943-t004:** Cement regulations following European Standard (EN 197-1) [59].

Raw Materials	Units	CEM I (Clinker)	CEM III (GGBFS)	CEM I + GP	CEM III + GP
Limestone	kg/kg product	1.1737	0.4049	0.5868	0.2025
Clay	kg/kg product	0.3307	0.1141	0.1653	0.0570
Sand	kg/kg product	0.0503	0.0174	0.0252	0.0087
Iron ore	kg/kg product	0.0203	0.0070	0.0102	0.0035
Water	m3/kg product	0.0668	0.0230	0.0334	0.0115
GGBFS	kg/kg product	-	1.0316	-	0.5158
Raw glass	kg/kg product	-	-	0.6480	0.6480

**Table 5 materials-16-00943-t005:** Energy consumption inventory input and calculations.

Energy	Units	CEM I (Clinker)	CEM III (GGBFS)	CEM I + GP	CEM III + GP
Electricity (kWh)	kWh/kg product	0.0687	0.0687	0.0403	0.0403
Petroleum coke	kg/kg product	0.0463	0.0463	0.0231	0.0231
Hard coal	kg/kg product	0.0559	0.0559	0.0280	0.0280
Diesel	kg/kg product	3.04 × 10^−7^	3.04 × 10^−7^	1.52 × 10^−7^	1.52 × 10^−7^
Fuel oil	kg/kg product	0.0117	0.0117	0.0058	0.0058
Natural Gas (MJ)	MJ/kg product	0.2777	0.2777	0.8768	0.8768
Light distribution	kg/kg product	0.0043	0.0043	0.0022	0.0022

**Table 6 materials-16-00943-t006:** General emissions inventory input and calculations.

Emissions	Units	CEM I (Clinker)	CEM III (GGBFS)	CEM I + GP	CEM III + GP
CO_2_	kg/kg product	0.0687	0.0687	0.0403	0.0403
NO_x_	kg/kg product	0.0463	0.0463	0.0231	0.0231
SO_2_	kg/kg product	0.0559	0.0559	0.0280	0.0280
H_2_O(g)	kg/kg product	3.04 × 10^−7^	3.04 × 10^−7^	1.52 × 10^−7^	1.52 × 10^−7^
Particulates	kg/kg product	0.0117	0.0117	0.0058	0.0058
Waste	kg/kg product	0.2777	0.2777	0.8768	0.8768

## Data Availability

The experimental and computational data presented in the present paper are available from the corresponding author upon request.

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
