# Peer review of "Insight into the Behavior of Mortars Containing Glass Powder: An Artificial Neural Network Analysis Approach to Classify the Hydration Modes"

_materials, 2023, doi:10.3390/ma16030943_

Round 1

Reviewer 1 Report

The introduction section is well written with well elucidated research significance. However, the ANN model is based on very limited experimental data from the authors and hence the  conclusions of the study are of limited reach.

Here are some comments for the authors to improve the quality of their paper:

1.      The choice of 50% of glass as cement replacement in the experimental tests  should be justified.

2.      CEMIII is of 32.4 strength class but it is finer than CEMI of 52.5 strength class. This should be checked.

3.      The authors used a lot “we can classify”,  “we used”,  “we proposed” , “we proceed” and so on. This should be avoided throughout the text by indirect style. “it was classified”, “It was done”, “it was proposed” it was proceeded” and so on.

4.      In the first sentence of the introduction, the statistics should be checked and compared to other references. With more than 4 billion tons of cement, concrete production should be more than 100 Mm3/year.

5.      There are some minor English mistakes. the paper should be checked thoroughly. Here are some examples. Line 117: “be beyond control” instead of “ be beyond of control“.

6.      Check the units: example lines 178 and 179 (3150 kg/m3 instead of 3150kg/m) and 2510 kg/m3 instead of 2510 kg/m3. 

7.      In figure 3, there seem to be an error in translation. it should be gypsum and not plaster (5%). There are other translations that should be checked as in table 6 where it should be “natural gas” and not “gas natural”.

Author Response

Materials 2141925

Insight into the Behavior of Mortars Containing Glass Powder: An Artificial Neural Network Analysis Approach to Classify the Hydration Modes

By: Fouad Boukhelf, Daniel Lira Lopes Targino, Mohammed Hichem Benzaama, Lucas Feitosa de Albuquerque Lima Babadopulos and Yassine El Mendili

Reply to Reviewers’ Comments

Dear Editor and Reviewers,

Thank you for reviewing our manuscript (Materials 2141925). We have considered the comments and revised the paper accordingly. Paragraphs added or corrected in the revised manuscript, according to reviewers’ comments, are marked up using the “Track Changes” function. The main improvements and the answers to the reviewers’ comments are given below:

Reviewer #1:

The introduction section is well written with well elucidated research significance. However, the ANN model is based on very limited experimental data from the authors and hence the conclusions of the study are of limited reach.

Here are some comments for the authors to improve the quality of their paper:

  1. The choice of 50% of glass as cement replacement in the experimental tests should be justified.
  2. CEMIII is of 32.4 strength class but it is finer than CEMI of 52.5 strength class. This should be checked.
  3. The authors used a lot “we can classify”, “we used”,  “we proposed” , “we proceed” and so on. This should be avoided throughout the text by indirect style. “it was classified”, “It was done”, “it was proposed” it was proceeded” and so on.
  4. In the first sentence of the introduction, the statistics should be checked and compared to other references. With more than 4 billion tons of cement, concrete production should be more than 100 Mm3/year.
  5. There are some minor English mistakes. the paper should be checked thoroughly. Here are some examples. Line 117: “be beyond control” instead of “ be beyond of control“.
  6. Check the units: example lines 178 and 179 (3150 kg/m3 instead of 3150kg/m) and 2510 kg/m3 instead of 2510 kg/m3.
  7. In figure 3, there seem to be an error in translation. it should be gypsum and not plaster (5%). There are other translations that should be checked as in table 6 where it should be “natural gas” and not “gas natural”.

Responses to reviewer #1:

The authors thank the reviewer for his encouraging comments. The responses to the relevant remarks are as follows:

Response to the overall remark: The authors thank you for this highlight.

In this particular case, the ANN model (MLP Regressor) was used for a specific case of input and output mapping function, in order to proceed with experimental data processing into a fitted model. The highlighted aspect mentioned should be highly observed for forecast methods, which is not the case. However, for the application into a mapping function, the adherence of the fitted model onto the experimental data can be assessed by its high R2 Score, followed by the low value of RMSE and MAE presented and, finally, in the “True vs Predicted” plots. Moreover, the amount of data is of course limited to the possibility of producing such data in laboratory in this rather new kind of application, but it should be noticed that 4 datasets (1 for each binder blend) was used and each dataset contains X data points, in a total of Y datapoints being analyzed. In the future, it is believed that models will be produced with more proficuous data, which is nowadays not possible but does not prevent the developments. Even forecast applications should be possible in the future.

Remark 1: The choice of 50% of glass as cement replacement in the experimental tests should be justified.

Response 1: Authors thank the reviewer for highlighting this point.

The objective of this study is to use the ANN model to predict the hydration process of the binder. To validate this model, it is more convenient to choose a binder with a low hydration rate. This is the reason why we have chosen a high substitution rate.

The following sentences are added to the manuscript. Please see section 2.1:

“Indeed, Zeybek et al [57] conducted a series of compressive strength, splitting tensile strength, and flexural strength tests to investigate the effect of waste glass powder on the mechanical properties of concrete. In their study, glass powder was used to replace up to 50% of cement. The study conducted by Kalakada et al [58] showed also a decrease in compressive strength at high cement substitution. The compression strength of the concrete specimen is reduced by 65% when 50% of the cement is replaced with glass powder. In this study, it was chosen to replace the cement by 50% of GP in mass in view of other experiences in the literature with high substitution rates, but it is important to notice that a focus is given on the proper validation of the relevance of the ANN model with materials having a low hydration heat (which is expected at those high substitution rates – up to 50%).”

Remark 2: CEMIII is of 32.4 strength class but it is finer than CEMI of 52.5 strength class. This should be checked.

Response 2: You are right, the cement CEM I is actually finer than CEM III. In fact, our results are obtained from a laser particle size analyzer by using the wet process.  Consulting the literature works and currently the studies of Kheir et al [53], it can be found that the CEM I is finer than the CEM III. In our study, the two cements used are very similar regarding the particle size distribution, but respected the trend of finer CEM I. Correction has been made to the text in Section 2.1.1.

Remark 3: The authors used a lot “we can classify”,  “we used”,  “we proposed” , “we proceed” and so on. This should be avoided throughout the text by indirect style. “it was classified”, “It was done”, “it was proposed”, "it was proceeded” and so on.

Response 3:

All sentences of the revised manuscript have been re-written in the indirect style. Thank you.

Remark 4: In the first sentence of the introduction, the statistics should be checked and compared to other references. With more than 4 billion tons of cement, concrete production should be more than 100 Mm3/year.

Response 4:

Thanks for the referred correction. You are right, based on Table 1 of reference [1], the production of cement is about 4 billion tons/year, while for concrete it fluctuates around 10 Gm3/year approximately. This mistake has been corrected in the revised manuscript. Please see lines 42-45.

Remark 5: There are some minor English mistakes. the paper should be checked thoroughly. Here are some examples. Line 117: “be beyond control” instead of “ be beyond of control“.

Response 5:  This mistake has been corrected in the final version and all the manuscript was checked for English mistakes, whose correction can be verified in the “Track Corrections” tool. Thank you.

Remark 6: Check the units: example lines 178 and 179 (3150 kg/m3 instead of 3150kg/m) and 2510 kg/m3 instead of 2510 kg/m3.

Response 6:  Thank you for this remark. Those mistakes have been corrected in the final version.

Remark 7: In figure 3, there seems to be an error in translation. it should be gypsum and not plaster (5%). There are other translations that should be checked as in table 6 where it should be “natural gas” and not “gas natural”.

Response 7:  Those two translation errors and others found in the paper after a thorough English revision have been corrected and Figure 3 has been modified and become Figure 1 following to the reviewer 4 recommendations.

Reviewer 2 Report

This manuscript evaluates the “Insight into the Behavior of Mortars Containing Glass Powder: An Artificial Neural Network Analysis Approach to Classify the Hydration Modes”. The manuscript is elaborately described and contextualized with the help of previous and present theoretical background. All the references cited are relevant to this area of research. The methods/analytical study are clearly stated. The result and discussion section are clearly presented. The manuscript needs Minor revision and require the following modifications before the acceptance.

1. What is the research need? Mention it in the abstract. Mention your research recommendation in the last line of abstract.

2. Cite the sentence ‘Indeed, concrete would be responsible for 4 to 8% of the world's CO2 emissions at all stages of  production with 50% of the emissions in the Construction sector alone’ needs citation. Some are found below.

https://doi.org/10.12989/acc.2021.12.4.327

https://doi.org/10.1002/suco.201800355

3. Mention your research gap.

4. Include more experimental photos

5. Compare your results with existing studies.

6. Present your research recommendations at the end of conclusion part.

Author Response

Materials 2141925

Insight into the Behavior of Mortars Containing Glass Powder: An Artificial Neural Network Analysis Approach to Classify the Hydration Modes

By: Fouad Boukhelf, Daniel Lira Lopes Targino, Mohammed Hichem Benzaama, Lucas Feitosa de Albuquerque Lima Babadopulos and Yassine El Mendili

Reply to Reviewers’ Comments

Dear Editor and Reviewers,

Thank you for reviewing our manuscript (Materials 2141925). We have considered the comments and revised the paper accordingly. Paragraphs added or corrected in the revised manuscript, according to reviewers’ comments, are marked up using the “Track Changes” function. The main improvements and the answers to the reviewers’ comments are given below:

Reviewer #2:

This manuscript evaluates the “Insight into the Behavior of Mortars Containing Glass Powder: An Artificial Neural Network Analysis Approach to Classify the Hydration Modes”. The manuscript is elaborately described and contextualized with the help of previous and present theoretical background. All the references cited are relevant to this area of research. The methods/analytical study are clearly stated. The result and discussion section are clearly presented. The manuscript needs Minor revision and require the following modifications before the acceptance.

  1. What is the research need? Mention it in the abstract. Mention your research recommendation in the last line of abstract.
  2. Cite the sentence ‘Indeed, concrete would be responsible for 4 to 8% of the world's CO2 emissions at all stages of  production with 50% of the emissions in the Construction sector alone’ needs citation. Some are found below.

https://doi.org/10.12989/acc.2021.12.4.327

https://doi.org/10.1002/suco.201800355

  1. Mention your research gap.
  2. Include more experimental photos
  3. Compare your results with existing studies.
  4. Present your research recommendations at the end of conclusion part.

Responses to reviewer #2:

The authors thank the reviewer for his encouraging comments. The paper has been improved accordingly to the responses to the relevant remarks, as indicated below:

Remark 1: What is the research need? Mention it in the abstract. Mention your research recommendation in the last line of abstract.

Response 1: We added the following sentence in the abstract “The proposed ANN model will allow cement manufacturers to quickly identify the different hydration modes of new binders by using only the heat hydration test as an input parameter.”

Remark 2: Cite the sentence ‘Indeed, concrete would be responsible for 4 to 8% of the world's CO2 emissions at all stages of  production with 50% of the emissions in the Construction sector alone’ needs citation. Some are found below.

https://doi.org/10.12989/acc.2021.12.4.327

https://doi.org/10.1002/suco.201800355

Response 2: Authors thank the reviewer for those two references defending the idea of the sentence. Those have been mentioned in the revised manuscript under reference numbers [3 and 4].

Remark 3: Mention your research gap.

Response 3: Authors added the last paragraph in the conclusion section for underlining the research gap.

Remark 4: Include more experimental photos

Response 4: Figure 1 was re-constructed and now almost all the tests have been mentioned in the conceptual plan.

Remark 5: Compare your results with existing studies.

Response 5:  The results were compared with those of the literature and in particular those resulting from the mechanical tests, porosity, ATG and some results of the ANN model. 

Remark 6: Present your research recommendations at the end of conclusion part.

Response 6:  We added the following sentence in the conclusion: “It was revealed in this study that waste glass can be employed as a partial cement substitute in mortar and concrete. The present study identifies waste glass as suitable, accessible in large quantities, a local eco-material, and suitable for concrete construction from an economic and environmental standpoint. Nevertheless, when waste glass was employed as a partial replacement for cement, the strength values were reduced. To address this shortcoming, future research will focus on combining waste glass with other recycling fibers. Moreover, the proposed ANN model allows cement manufacturers to quickly identify the different hydration modes of new binders by using only the heat of hydration test as an input parameter.”

Reviewer 3 Report

Dear Author.   Congratulations on the work you have done.   In this paper, the properties of mortar containing glass powder are investigated and artificial neural network model is used to predict its hydration process.   Introduction. The introduction is at an adequate level and it explains all the necessary information for the next section. Materials and methods. In this section, certain paragraphs can be improved (see Problem). Results and discussion. The text section is sufficiently detailed. Conclusion. The conclusion section briefly summarizes the results of the study. The list of publications used is sufficient.    In general, the structure and content of the manuscript is acceptable for the material. Nevertheless, to improve its readability, please consider the following suggestions.   1 Some data results can be presented in the abstract.   2 Please explain the basis for the choice of plasticizer and the reasons for the variation in content.   3 In the experimental protocol section, more details could be added to describe the equipment and test protocol.   4 It is suggested to sort out the results to clarify the findings and innovations.

Author Response

Materials 2141925

Insight into the Behavior of Mortars Containing Glass Powder: An Artificial Neural Network Analysis Approach to Classify the Hydration Modes

By: Fouad Boukhelf, Daniel Lira Lopes Targino, Mohammed Hichem Benzaama, Lucas Feitosa de Albuquerque Lima Babadopulos and Yassine El Mendili

Reply to Reviewers’ Comments

Dear Editor and Reviewers,

Thank you for reviewing our manuscript (Materials 2141925). We have considered the comments and revised the paper accordingly. Paragraphs added or corrected in the revised manuscript, according to reviewers’ comments, are marked up using the “Track Changes” function. The main improvements and the answers to the reviewers’ comments are given below:

Reviewer #3:

Dear Author.   Congratulations on the work you have done.   In this paper, the properties of mortar containing glass powder are investigated and an artificial neural network model is used to predict its hydration process.   Introduction. The introduction is at an adequate level and it explains all the necessary information for the next section. Materials and methods. In this section, certain paragraphs can be improved (see Problem). Results and discussion. The text section is sufficiently detailed. Conclusion. The conclusion section briefly summarizes the results of the study. The list of publications used is sufficient.    In general, the structure and content of the manuscript is acceptable for the material. Nevertheless, to improve its readability, please consider the following suggestions.   1 Some data results can be presented in the abstract.   2 Please explain the basis for the choice of plasticizer and the reasons for the variation in content.   3 In the experimental protocol section, more details could be added to describe the equipment and test protocol.   4 It is suggested to sort out the results to clarify the findings and innovations.

Responses to reviewer #3:

The authors thank the reviewer for his encouraging comments. The responses to the relevant

remarks are as follows:

Remark 1: 1 Some data results can be presented in the abstract.  

Response 1: The abstract is reworked to answer the reviewer's remark. Thank you.

Remark 2: Please explain the basis for the choice of plasticizer and the reasons for the variation in content.  

Response 2: We added this sentence in the mix design section. “A SikaCem® superplasticizer polycarboxylate was added to all the mixtures in order to obtain a better and comparable consistency. Indeed, due to the low content of the superplasticizer, compared to the mortar constituents, its content is neglected”.

Remark 3: In the experimental protocol section, more details could be added to describe the equipment and test protocol.  

Response 3: Authors thanks the reviewer for highlighting this remark. For this, section 2.1.2 has been reworked in the final version.

Remark 4: It is suggested to sort out the results to clarify the findings and innovations.

Response 4:  Authors thank the reviewer for highlighting this point. We added this sentence in the conclusion: “It was revealed in this study that waste glass can be employed as a partial cement substitute in mortar and concrete. The present study identifies waste glass as suitable, accessible in large quantities, a local eco-material, and suitable for concrete construction from an economic and environmental standpoint. Nevertheless, when waste glass was employed as a partial replacement for cement, the strength values were reduced. To address this shortcoming, future research will focus on combining waste glass with other recycling fibers. Moreover, the proposed ANN model allows cement manufacturers to quickly identify the different hydration modes of new binders by using only the heat of hydration test as an input parameter”.

Reviewer 4 Report

Dear Authors,

please find my comments:

1. Line 460 - Figure 13. Heat hydration, ANN model generated, modes and transitions identification: (a) CEM I and (b) CEM I + GP. And Line 463 - Figure 13. Heat hydration, ANN model generated, modes and transitions identification: (a) CEM III and (b) CEM III + GP. Have the same number !

2. Line 198 - Table 2. Mix proportions per cubic meter of mortar (kg/m3 ). The table name confirms what is in parentheses. Or the name should be left, and if not, change it to - Mix proportions of mortar (kg/m3)

3. What kind of superplasticizer was used? And how much of it was used from the amount of cement?

4. I think it would be more logical to move section 2.1.3 Life Cycle Analysis LCA to Introduction part.

5. I recommend combining part 2.1.4 Experimental protocol with 2.1.2 Mix design. Because the composition of the mortars is followed by the preparation and curing of the samples.

6. Line 32 - Regarding GP production, its manufacturing process shown in Table S1 is not on an industrial scale according to [60]. Table S1 is not available.

Author Response

Materials 2141925

Insight into the Behavior of Mortars Containing Glass Powder: An Artificial Neural Network Analysis Approach to Classify the Hydration Modes

By: Fouad Boukhelf, Daniel Lira Lopes Targino, Mohammed Hichem Benzaama, Lucas Feitosa de Albuquerque Lima Babadopulos and Yassine El Mendili

Reply to Reviewers’ Comments

Dear Editor and Reviewers,

Thank you for reviewing our manuscript (Materials 2141925). We have considered the comments and revised the paper accordingly. Paragraphs added or corrected in the revised manuscript, according to reviewers’ comments, are marked up using the “Track Changes” function. The main improvements and the answers to the reviewers’ comments are given below:

Reviewer #4:

Dear Authors,

please find my comments:

  1. Line 460 - Figure 13. Heat hydration, ANN model generated, modes and transitions identification: (a) CEM I and (b) CEM I + GP. And Line 463 - Figure 13. Heat hydration, ANN model generated, modes and transitions identification: (a) CEM III and (b) CEM III + GP. Have the same number!
  2. Line 198 - Table 2. Mix proportions per cubic meter of mortar (kg/m3 ). The table name confirms what is in parentheses. Or the name should be left, and if not, change it to - Mix proportions of mortar (kg/m3)
  3. What kind of superplasticizer was used? And how much of it was used from the amount of cement?
  4. I think it would be more logical to move section 2.1.3 Life Cycle Analysis LCA to Introduction part.
  5. I recommend combining part 2.1.4 Experimental protocol with 2.1.2 Mix design. Because the composition of the mortars is followed by the preparation and curing of the samples.
  6. Line 32 - Regarding GP production, its manufacturing process shown in Table S1is not on an industrial scale according to [60]. Table S1is not available.

Responses to reviewer #4:

The authors thank the reviewer for his encouraging comments. The responses to the relevant remarks are as follows:

Remark 1: Line 460 - Figure 13. Heat hydration, ANN model generated, modes and transitions identification: (a) CEM I and (b) CEM I + GP. And Line 463 - Figure 13. Heat hydration, ANN model generated, modes and transitions identification: (a) CEM III and (b) CEM III + GP. Have the same number!

Response 1: Actually, the last figure is under the number 14. This mistake has been corrected in the final version. Thank you for the remark.

Remark 2: Line 198 - Table 2. Mix proportions per cubic meter of mortar (kg/m3). The table name confirms what is in parentheses. Or the name should be left, and if not, change it to - Mix proportions of mortar (kg/m3)

Response 2: Authors thank the reviewer for this remark. It was chosen “Mix proportions of mortar (kg/m3)” for the Table 2 title.

Remark 3: What kind of superplasticizer was used? And how much of it was used from the amount of cement?

Response 3: We added this sentence in the mix design section. “A SikaCem® superplasticizer polycarboxylate was added to all the mixtures in order to obtain a better and comparable consistency. Indeed, due to the low content of the superplasticizer, compared to the mortar constituents, its content is neglected. The superplasticizer/cement ratio varies between 0.36% and 0.54%.”.

Remark 4: I think it would be more logical to move section 2.1.3 Life Cycle Analysis LCA to Introduction part.

Response 4: Section 2.1.3 of the old version has been moved to the Introduction section of the final section. We do think this improved the paper. Thank you. 

Remark 5: I recommend combining part 2.1.4 Experimental protocol with 2.1.2 Mix design. Because the composition of the mortars is followed by the preparation and curing of the samples.

Response 5: Sections 2.1.2 and 2.1.4 have been combined in the final version. Thank you.

Remark 6: Line 32 - Regarding GP production, its manufacturing process shown in Table S1 is not on an industrial scale according to [60]. Table S1 is not available.

Response 6: Table S1 is actually Table 4 in the final version. All Tables and their citations were updated in the final version. 
